# Prevalence of Plastic and Hardware Foreign Bodies among Goats at Malawi Markets

**DOI:** 10.3390/ani14010147

**Published:** 2024-01-01

**Authors:** Paul M. Airs, Jonathan H. I. Tinsley, Winchester Mvula, Javier Ventura-Cordero, Taro Takahashi, Patson Nalivata, Jan A. van Wyk, Eric R. Morgan, Andrews C. L. Safalaoh

**Affiliations:** 1Biological Sciences, Queen’s University of Belfast, 19 Chlorine Gardens, Belfast BT9 5DL, UK; jtinsley03@qub.ac.uk (J.H.I.T.); venti19@hotmail.com (J.V.-C.); eric.morgan@qub.ac.uk (E.R.M.); 2Animal Science Department, Lilongwe University of Agriculture and Natural Resources (LUANAR), Lilongwe P.O. Box 219, Malawipatienalivata@yahoo.com (P.N.);; 3Agri-Food and Biosciences Institute, Hillsborough, Co. Down, Northern Ireland BT16 6DR, UK; 4Department of Veterinary Tropical Diseases, University of Pretoria, Pretoria Private Bag X20, South Africa

**Keywords:** small ruminants, plastic, pollution, butcher, animal health, indigestible foreign bodies, smallholder

## Abstract

**Simple Summary:**

Goat smallholdings are common in both rural and urban areas of Malawi, and are kept as a food source and as investments to improve livelihoods. However, goats are known to forage on indigestible items that can negatively impact health and lead to a loss of productive performance. To date, no studies have estimated the prevalence of pollution among Malawi goats despite issues with waste management. To this end, we sought to survey the frequency of plastic and other indigestible foreign bodies among goats slaughtered by informal market butcher stands in five districts spanning Malawi. Most of the butchers surveyed identified plastic during slaughter (80%) while almost half (45.3%) identified other indigestible objects (hardware). Plastic was found by butchers in all districts and across rural and urban settings. While being less common than plastic, the hardware noted by butchers included sharps such as needles and bicycle spokes. When purchasing, butchers do consider animal health important, but 70.7% consider injury status as less or not important. Overall, this study highlights the issues of pollution among smallholder goats and demonstrates the need for further study to address the impacts of pollution on animal health and knock-on impacts on smallholder livelihoods.

**Abstract:**

Smallholder goat production plays a major role in rural livelihoods and food security in Malawi, but suffers from drastic and unpredictable production losses. While goat production is closely linked to small-scale local markets for slaughter and butchering, the perspectives of butchers and their potential as a source of animal health information are largely untapped. Butchers can provide insights into goat health status at slaughter as well as issues that go unseen before slaughter, such as the presence of indigestible foreign bodies (IFBs). IFBs include solid materials such as plastics and hardware (metals, stones, and other hard objects) that cause foreign body syndrome and can lead to impaction, oedema, malnutrition, and death. To estimate the presence of IFBs, 150 market stand butchers were surveyed across five districts in Malawi, focusing on a distinction between hardware and single-use plastics, which are still widely present in Malawi despite bans on production. Most butchers found plastic IFBs (80.7%), with over half (56.7%) reporting plastic IFBs recently among the past five slaughters. Hardware IFBs were less common, reported by 45.3% of butchers. While some butchers commented on the impact of IFBs on meat quality metrics ex-post, the majority observed no differences. While butchers unanimously considered health to be an important characteristic when sourcing goats, 70.7% consider injury status to be less important or not important. Overall, this study highlights the issue of anthropogenic waste pollution on goat production in Malawi and demonstrates the potential for the surveillance of goat health at market.

## 1. Introduction

Goats function as a living asset in Malawi and many other agrarian subsistence cultures worldwide. In Malawi, goats are the most common smallholder livestock kept, after chickens, and act as an investment which can rapidly be exchanged for cash when needed [1,2,3]. Attention to goat production, such as through supplementation, effective breeding, and the prevention of diseases, is linked to livelihood improvements among smallholders in Malawi [1,4,5,6]. However, Malawi smallholdings frequently suffer from livestock losses that result in net losses to flock size [6], with ~20% of losses due to unknown causes [7]. Since plastic and other solid waste pollution is common in Malawi with thousands of tonnes generated daily [8], we posit that anthropogenic debris may impact goat health and production among Malawi smallholdings.

Plastics and other anthropogenic debris seriously threaten the future of food security in a number of sectors including human health [9,10], soil quality and agricultural yield [11,12], and production and management at fisheries [10,13]. In livestock, solid debris aggregates as indigestible foreign bodies (IFBs) following ingestion, most commonly from plastic, but also from paper, rope, cloth, leather, caked sand and seeds, maize kernels, metallic objects, and stones [14,15,16,17,18,19,20]. In small ruminants, the ingestion of IFBs has a number of negative impacts that are known collectively as ‘foreign body syndrome’, which can lead to gut impaction, haemorrhages, rumenitis, ruminal acidosis, and anaemia [16,21,22,23]. While clinical consequences range from inapparent to lethal, IFBs are generally detrimental to body condition [16,22,24] and are correlated to lower weight, poor body condition, and anaemia [15,16,19,23,25]. Among small ruminants, goats are more likely to suffer from conditions such as the oedema of the lamina propria compared to sheep [23]. In Kenya, IFBs caused by plastic pollution, especially single-use plastic bags, have been associated with goat illness and death [26], but whether these are identified at market in rural smallholder settings needs further study. Further, while IFBs have been documented in Nigeria [16,20], Sudan [27], Ethiopia [15,19,28], Kenya [14,26], and Tanzania [17], there is a lack of data for Malawi.

The visible impacts of IFBs mirror conditions from nutritional deficiency and infections with gastrointestinal nematodes and other helminth parasites, which are often diagnosed by poor body condition, anaemia, oedema, and other pathophysiological indicators [29]. As such, IFBs may also conflate signs used for the monitoring of parasites and productive performance and require slaughter for accurate detection. Monitoring foreign body syndrome visually is also complicated due to the low resources available, limited access to veterinary care, and lack of abattoir settings for smallholder goat slaughter. Since market butchers are the principal individuals witnessing otherwise invisible health issues during smallholder goat slaughter, engagement with butchers could lead to the documentation and dissemination of knowledge relating to goat health back to veterinary services and smallholders. Although training would be necessary to accurately determine the presence of other health factors, the presence of IFBs and overall goat characteristics are visible to butchers during slaughter. This is especially relevant in Malawi, where all parts of slaughtered animals are utilised, including intestinal tracts. As such, we sought to gain insight into the prevalence of visible traits among smallholder goats through surveying market butchers in both urban and rural settings. The terminology ‘market butcher’ employed here describes the interviewees, all of whom were value chain actors who slaughter, process, and market goat-derived foodstuffs resulting from slaughters. 

The aims of this study are twofold. First, we aim to investigate the prevalence and nature of IFBs across five districts of Malawi. Secondly, we aim to assess the knowledge and awareness of IFBs by market butchers, in addition to gathering their perceptions of impacted meat quality metrics. We hypothesise that IFBs are widely prevalent and of diverse form across Malawi, and that their impact is currently underestimated by those involved in the value chain. 

## 2. Materials and Methods

### 2.1. Study Sites and Recruitment

Social distancing was maintained for the collection of interview responses following the Malawi government guidelines relating to COVID-19, and ethical approval for the survey was granted by the Lilongwe University of Agriculture and Natural Resources. Market butchers were recruited through purposive and snowballing sampling methods. To be eligible to take part in the study, potential participants had to have slaughtered at least one goat in the previous year. The questionnaire was developed by the research team alongside local agricultural NGO workers and a retired meat quality inspector. Enumerators travelled to participants and invited and undertook interviews in-person. Verbal consent was granted by respondents who were informed about the purpose of the study, possibility for re-visitation, and publication of the data collected. Respondents were also informed that they were not obliged to take part in the study, could withdraw their participation at any time, and that responses would be fully anonymised.

A total of 150 respondent butchers were surveyed, with an initial 50 respondents in the Lilongwe district from July 2020 and an additional 100 respondents from April to July 2023 in the Chitipa, Nsanje, Salima, and Thyolo districts across Malawi (Figure 1a). The sample size of 150 was agreed by the research team as manageable given the limited time and financial resources. Although purposive and snowball sampling can lead to selection bias, the small number of butcheries in rural areas in Malawi meant that in many cases, no potential participants were excluded from the study. The discrepancy in time was due to the restrictions relating to the ongoing COVID-19 pandemic preventing survey collection. Each market visited was designated as a city/town, rural town, or village, defined by whether the estimated population of the location was 10,000 people or above for a city/town, 2500–9999 for a rural town, or below 2500 for a village. 

Images were collected with consent from butchers and local authorities in the Lilongwe (October 2019) and Chitipa (March 2023) districts for illustrative purposes (Figure 1b–d).

### 2.2. Survey Design

The questionnaire was divided into several sections: (1) Demographic information pertaining to the market location, butcher age, sex, years of experience, and inspections from animal health personnel, (2) Goat sourcing information including where and from whom goats are bought from, (3) Goat characteristics including sex, age, and health status, (4) The importance of goat characteristics in purchasing, (5) The periodicity of sales throughout the seasons, (6) Presence and issues relating to IFBs and dog bites. Following local terminology, IFBs were divided into two categories: plastic and ‘hardware’, comprising any other indigestible material. Additional questions were present in the questionnaires but are excluded from this analysis as they are part of a wider study on butcher market chains, irrelevant to this study. All anonymised questionnaire responses are available in Appendix A.

To determine variability in the answers provided by respondents, 50 respondents from Nsanje were revisited 15–19 days following the initial survey and asked a subset of questions (see Appendix A). The additional comments made by butchers detailing the types of hardware retrieved were made openly by butchers but recorded down by enumerators. Later surveys also invited butchers’ comments on the impact of plastic IFBs on meat quality to indirectly assess IFB impacts on health.

### 2.3. Statistical Analyses

Survey responses were tabulated in an Excel spreadsheet (Microsoft Inc., Redmond, Washington, DC, USA), with statistical analyses and graphical representations generated in GraphPad Prism version 9.0.0 for Windows (GraphPad Software, San Diego, CA, USA). Maps were produced using ArcGIS Pro version 3.2 (Environmental Systems Research Institute (ESRI), Redlands, CA, USA).

## 3. Results

Anthropogenic waste, commonly witnessed in grazing areas and in open refuse areas in many developing countries, was noted in study areas (Figure 1c,d). To determine the prevalence and spread of IFBs among goats brought to market in Malawi, we assessed 150 butchers in slaughterhouses across five districts.

### 3.1. Descriptive Statistics of Butchers and Their Businesses

All butchers interviewed were male (*n* = 150/150) business owners (*n* = 98/98) operating informal abattoirs (as shown in Figure 1b), with an average experience of 10 years across the entire sample (Appendix A). Businesses were small, with 45% having no employees, 28% having one employee, and 27% having more than one employee or seasonal help. When asked about inspections by animal health personnel, 47% of participants sample-wide responded ‘yes’, with a combined 53% saying ‘no’ (21%) or ‘pass’ (32%) (Appendix A).

To link goat sales to particular times, butchers were asked to rank sales throughout the year. Sales occurred throughout the year, even during the middle of the dry season where sales were reported as ‘rare’ or ‘lowest’. However, sales varied with the majority of butchers reporting the highest sales during festivities (88%) and after harvest (66%) (Appendix A).

### 3.2. Goat Sourcing

Goats were primarily bought from males (*n* = 140/149), who were the owners of the goats (*n* = 93/99), or middlemen (*n* = 6/99) and were sourced from both local and distant locations. In total, 49% of butchers sourced all goats from local areas, 44% from a mixture of local and distant areas, and 7% from distant areas exclusively (Appendix A). Butchers generally did not have a preference for local goats, responding that the distance from the market is ‘not important’ (32.7%) or ‘less important’ (50.3%) (Appendix A). The distances that goats travel to the market was not measurable and as such, the areas in which goats graze were not assessed directly. In addition to distance, butchers considered theft to be important and valued knowledge of theft as ‘very important’ (66.9%) or ‘less important’ (31.75%) (Appendix A).

### 3.3. Characteristics of Goats Purchased by Butchers

Butchers were asked to rank their preferences and frequencies of different characteristics in goat purchase decision making (Figure 2). Size and health were the most important factors (Figure 2a,b). Both male and female goats were commonly brought to market (ratio 10:9, no significant differences by Chi-square males vs. females, *p* = 0.1048), while old goats were more commonly brought to market and young goats were rarely seen (Chi-square of young (0 to 2 years old) vs. old (2+ years old), *p* < 0.0001). Unsurprisingly, injured and sick goats were reported as the rarest or never purchased by butchers (Figure 2a). However, injury status is considered to be ‘less important’ or ‘not important’ by the majority of butchers (Figure 2b). Contrastingly, healthy fat goats were unanimously reported as the most common (*n* = 149/150), with health being ‘very important’ for all respondents (*n* = 149/150) and size being ‘very important’ to the majority (*n* = 140/150) (Figure 2a,b).

### 3.4. Estimating the Prevalence and Distribution of Plastic and Hardware IFBs

Butchers were asked whether plastic or hardware was generally found in their goats with response options including: ‘no’, ‘yes—not common’, and ‘yes—major problem’ (Table 1). Plastic was more frequently noted compared to hardware (80.7% vs. 45.3%, Fisher’s exact test, *p* < 0.001), but most butchers did not consider plastic to be a ‘major problem’ (10%). Overall plastic was found across districts with 64–100% of respondents noting plastic to be present in goats, while hardware was more variable (0–76%). The Thyolo district reported no hardware IFBs at all, however only 12 butchers were available to interview in this region.

The markets visited were classified as being in a ‘city or town’ (*n* = 22), ‘rural town’ (*n* = 80), or ‘village’ (*n* = 46). Overall, more butchers in rural towns reported IFBs for both plastic (Chi-square, *p* < 0.0001) and hardware (Chi-square, <0.0001) compared to villages and city/town locations (Appendix A). However, these classifications did not yield any differences in the amount of plastic or hardware IFBs identified among recent slaughters (Appendix A) with both plastic (Chi-square of ‘any’, *p* = 0.3845) and hardware (Chi-square of ‘any’, *p* = 0.7966) equally distributed across location types.

### 3.5. Plastic and Hardware IFBs among Recent Slaughters

To gain an estimate of IFB frequency in lieu of abattoir settings, butchers were asked to count the number of goats containing plastic or hardware IFBs from the five most recent slaughters. Overall, 56.7% of butchers reported plastic and 25.7% of butchers reported hardware in at least 1/5 of the most recent goats slaughtered (Table 2). Using this metric, a total of 25.6% of most recent slaughters were reported with plastic IFBs, averaging at 1.3 goats in every 5. Comparatively, 7.3% of goats contained hardware, averaging 0.4 goat in every 5. The findings varied somewhat by location, with the Lilongwe district containing the highest burden of plastic among recently slaughtered goats, while no differences were identified between any other districts (Kruskal-Wallis test with Dunn’s multiple comparisons test, *p* < 0.001 for Lilongwe vs. all other locations only). The same was true for hardware IFBs with Lilongwe higher than all other districts (Kruskal-Wallis test with Dunn’s multiple comparisons test, *p* < 0.001 for Lilongwe vs. all other locations only). Comparatively, the Nsanje district reported the lowest overall plastic IFBs (10% of goats) and Thyolo the lowest hardware (0%) among recent slaughters.

To measure variability between visits as a means of determining stochasticity in butchers’ opinions alongside the presence of IFBs, all 50 of the Nsanje district respondents were revisited from 15 to 19 days following the initial questionnaire for a follow-up. Overall, some respondents changed their opinion of whether plastic (17/50 different) or hardware (16/50 different) was found in their slaughters, but opinions were not statistically significant, with the majority unchanged (Two-tailed McNemar’s test with continuity correction, *p* = 1.0 for plastic, and *p* = 0.3320 for hardware, see Appendix A). Among recent slaughters, plastic IFBs were found by more butchers in the second survey (*n* = 12/50 vs. 23/50), but the frequency of plastic did not change (*n* = 25 vs. 30), averaging at 0.5 instances per five goats in the first survey and 0.65 in the second (Appendix A). Comparatively, individual goats with IFBs (among the most recent slaughters) were significantly different for both plastic and hardware (Two-tailed McNemar’s test with continuity correction, *p* = 0.0164 for plastic, and *p* = 0.0244 for hardware, see Appendix A). These results indicate that the overall frequency of IFBs did not change and the presence of plastic and hardware within individual goats slaughtered is sporadic, as would be expected. However, the increased number of respondents witnessing IFBs in the second survey is possibly a sign of increased awareness.

A subset of butchers (*n* = 50) included details of the hardware retrieved, including needles (*n* = 22), nails (*n* = 2), pins (*n* = 5), and bicycle spokes (*n* = 8), as well as keys (*n* = 1), and coins (*n* = 12). The presence of such sharp items poses a health risk to goats as well as to butchers and consumers. A different subset of butchers (*n* = 93) were asked whether plastic or hardware IFBs impacted slaughter. Most butchers did not notice any changes (72%), but those that did demonstrated health impacts following IFB consumption (Table 3). The most common comments were related to a lack of fat on the carcass directly or indirectly through weight loss or changes to meat quality.

## 4. Discussion

In this study, we aimed to profile the distribution and scale of IFBs among slaughtered goats at Malawi markets across the country. Plastics were found to be extremely common, found by 80.7% of all butchers and in 25.6% of all recently slaughtered goats. Plastic IFBs far outweighed hardware in terms of prevalence and spread, which is unsurprising since a recent cleanup effort on the shores of Lake Malawi revealed that 80% of the anthropogenic debris comprised plastic litter, with plastic carrier bags being the most common item [30]. Plastic IFBs, especially plastic bags, are also the most common form extracted from ruminants in developing countries [24] and are common across Africa [14,15,16,17,18,28]. 

The high proportion of IFBs identified could be due to the preference for older, larger goats at Malawi markets (see Figure 2). IFBs are known to accumulate over time in ruminants [14,25,28], and as such, market selection may highlight the extremity of plastic pollution. However, it is also possible that the consumption of IFBs leads to malnourishment, which reduces performance and causes goats to take more time to reach a desirable size [24]. Conversely, malnourishment could drive the ingestion of IFBs. Both possibilities are further discussed below. Furthermore, improved awareness of IFBs by butchers could implicate an enhanced awareness of their prevalence and impact; participants in Nsanje were interviewed twice, where the perceived prevalence increased between the first and second interviews. While this could be evidence of response bias, it is likely that education on IFBs and subsequently an enhanced awareness accounts for some of the variation and imply that IFBs are more prevalent than initially recognised by participants. 

Other than plastic, the prevalence of hardware was surprisingly high, with sharp items recovered by butchers including needles, pins, nails, and bicycle spokes. In one Kenyan study, metal wires were present in 4.5% of goats [14], while another study in Ethiopia reported metal in 0.9% of ruminants [19]. Here, hardware overall was present in 7.3% of goats, which is high, but not far out of step with other studies. Metals and other hardware are reported in small ruminant IFB studies but are more common in the less-discriminate bovids compared to goats and sheep [14,15,16,17,18,19,20]. As such, the uptake of these debris in goats may be due to starvation, excessive pollution, or both, as goats in the same areas tend to have less hardware identified compared to other ruminants [15,18].

If the costs of continually cleaning anthropogenic debris are too great, there may be a need to switch to biodegradable plastics. A recent study in cattle identified that some forms of biodegradable plastics can be digested, reducing the risk of solid obstruction formation [31]. While this situation is far from ideal, and reusable packaging and bags are strongly encouraged by the U.N. and others [8,30,32], it is evident that some IFBs are more problematic than others, and could become focal points. In this case, single-use plastic bags are the most common and most problematic anthropogenic debris across a number of sectors [11,12].

Despite the high reported prevalence of IFBs, it appeared that most butchers were not overly concerned by plastic and other anthropogenic waste impacting animal health and meat quality as none claimed hardware to be a ‘major problem’ and only 10% said the same of plastic. Although most butchers did not comment on the impact of IFBs on meat quality or claim no difference, those who made comments appear to indicate that IFBs are a result of poor nutrition. The comments reference a lack of fat, reduced weight, or the toughness of meat, which can be a result of emaciation from IFB impaction (see Table 2). This echoes the findings from others where IFBs are highly correlated to poor body condition [15,16,22,28]. Conversely, prior nutritional deficiency could also drive goats to consume IFBs, especially those which housed foodstuffs and contain traces of salt, sugar, or oils, as has been anecdotally witnessed [33]. Farmers in Kenya also reported that a lack of feed was driving the consumption of plastic [26]. 

Given that IFBs can cause impaction, haemorrhage, rumenitis, and ruminal acidosis, which lead to anaemia and emaciation, it seems likely that IFBs are both a contributing factor to poor body condition and exacerbate prior nutritional deficiency [16,21,22,23]. In Sudan and Ethiopia, high levels of IFBs are reported and make up ~50% of goat surgical interventions [27,34]. As such, it is possible that a lack of awareness exists relating to the damage caused by IFBs in Malawi, and with it a real and urgent need to educate butchers as sentinels to detect when and where IFBs may be leading to the unnecessary and premature slaughter of goats. This is also bearing in mind that IFBs are only monitored from goats that reach slaughter and do not account for sporadic and unexplained deaths.

However, the conditions and risk factors that drive IFB uptake in goats remain to be fully explained. In some areas, grazing on refuse dump areas yields obvious answers due to poor forage and excessive pollution [16,25,34]. The increased prevalence of IFBs in the urban-dwelling goats of Nigeria, which have lower body condition scores compared to rural goats, points to the access to anthropogenic debris alongside nutrient deficit being co-drivers of IFBs [25]. In the present study, the frequency of IFB detection varied from location to location, being more common in Lilongwe (see Table 1), the most densely populated area of Malawi. Since we had limited resources for collecting responses and due to the restrictions owing to the COVID-19 pandemic, the variability of IFBs among goats seasonally and across years could not be determined. Previously, we demonstrated that smallholders in central Malawi tether goats during the growing season and provide supplementation, and free graze post-harvest [6,35]. It is likely that these practices lead to times of increased risk for the consumption of IFBs, such as when goats are tethered at the start of the growing season following the first rains, before sufficient supplementation is available and forage in common grazing areas is limited. It is known that tethering without proper supplementation leads to poor goat nutrition [36,37], reduces growth, and can subsequently reduce goat value at market [1,38,39]. These limitations are compounded by seasonal gastrointestinal nematode parasite burdens that drastically impact the body condition and survival of goats, unless controlled effectively [39,40,41,42]. Since IFBs are known to be problematic in times of nutritional deficiency or in conditions where seasonal rains impact forage availability [24] and increase parasite burdens [43], knowing the timing and situations when goats are more likely to selectively ingest IFBs will be essential in improving goat productive performance and heath, and warrants further study.

Understanding the interplay between IFB ingestion and the other stressors induced on goat health may lead to policies and efforts that maximise the chances of livestock productive performance and survival. For instance, intervention strategies could focus on plastic removal at key points in the year when goats are suffering from nutritional deficit and are more likely to selectively browse on plastic. It could also be that, when provided with improved supplementation, goats will avoid IFBs entirely, as has been suggested by farmers who continue to rely on single-use plastics due to convenience and a lack of other options [26]. Regardless of the possible outcomes, the information disconnect between the veterinary service, farmer, and butcher can negatively impact the livelihoods of all parties, since butchers may witness the causes of illness during slaughter but not relay this information back to veterinary personnel or the farmer. If farmers and local veterinary personnel could be made aware of the presence of IFBs and any correlations with pathology following slaughter, the scope and impact of IFBs can be properly addressed. This feedback is critical since slaughter may be a result of the poor performance caused by IFBs. Witnessing high levels of IFBs by butchers could also trigger broader efforts to reduce pollution, which was the case for Gloria Majiga-Kamoto, whose witnessing of goats consuming plastic led to campaigns for single-use plastic bans in Malawi [33]. However, there is a need to both enforce these bans effectively and install improved waste collection infrastructure [24,32]. Given that an estimated 60,000 tonnes of single-use plastic per year are produced [8,32], with thousands of tonnes of solid waste generated every day just in Malawi’s urban centres [8], the prevalence of IFBs could also be used to measure pollution levels and prompt cleanup efforts. Cleanup efforts have been made in Malawi but are not yet a systemic part of the governmental policy, despite such effort yielding an improvement of the land and water systems [30].

Ultimately, systemic interventions are needed to ensure goat safety and the control of ownership. A sustainable and incentivised approach to control refuse pollution in Malawi is desperately needed as refuse undoubtedly has a negative impact on many other livestock and other aspects of rural life and wellbeing, including human health [10,11,12]. There are calls for the elimination of single-use plastic in Malawi [8], and IFB research heeds a similar message to invest in better waste management systems [17,28]. Overall, further study of butcher perspectives, the reasons for goat slaughter, and the impact of IFBs in livestock is needed to ensure the stability of livelihoods in smallholder economies.

The limitations of this study should be highlighted. Firstly, the purposive and snowball sampling methods utilised in this study are known to risk selection bias. However, efforts were made to identify and include all butchers in each survey location to mitigate this risk. Next, response bias could have affected the results, as the verbal consent process highlighted that the purpose of the study was to investigate the presence of IFBs. Yet, while butchers may have consequently overstated their commonness, given the implications of IFBs on meat quality, it is also possible that butchers underplayed the prevalence and impact of IFBs to safeguard their reputations, which may explain the disparity between the high prevalence observed by butchers and their perception of the impact on the meat quality metrics displayed within Table 3. To account for these drawbacks and further explore the factors relating to the prevalence and impact of IFBs, future research should be undertaken. We suggest three options; firstly, a longitudinal participant observation study of slaughtering facilities would help to elucidate potential spatiotemporal fluctuations in IFB prevalence. Secondly, the effect of different concentrations of IFB pollution on the ingestion rate would help to develop guidelines and monitoring metrics on risk thresholds. Social network analysis could be employed to identify the farms where goats originate from, at which point surveys could be undertaken to assess the concentration of IFBs in the associated grazing areas and the observed IFB concentrations in slaughtered goats. Lastly, a study investigating the impact of different nutritional availability contexts and goat grazing behaviour would help to understand if goats are at higher risk of ingesting IFBs during episodes of nutritional deficit and poor browse availability, such as during droughts, or in cases of poorly managed tethering methodologies. 

## 5. Conclusions

We find that IFBs, principally plastics, are widespread among the smallholder goats brought to slaughter in Malawi. The IFBs identified by butchers also included sharp metal objects likely to cause haemorrhages and damage to internal organs as well as luminal impaction. While the majority of butchers believe IFBs not to be a major problem, some correlated IFB presence with emaciation. Overall, there is limited knowledge of IFB impact on goats among butchers. If knowledge transfer relating to IFBs can be generated between butchers, farmers, and local veterinary personnel, the scope and impact of IFBs can be properly addressed and the true impact of IFBs on goat health and human livelihoods can be determined.

## Figures and Tables

**Figure 1 animals-14-00147-f001:**
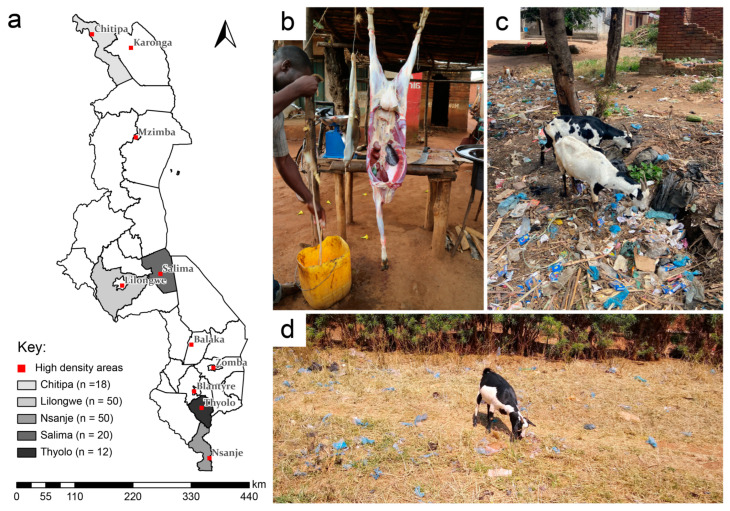
Study area and representative images. (**a**) Map of study area with study districts and number of respondents listed. (**b**–**d**) Representative images including, (**b**) a typical informal market butcher stand during goat slaughter and cleaning of intestine, (**c**) goats free grazing in a refuse dump area, and (**d**) a goat free grazing in a village amongst anthropogenic debris.

**Figure 2 animals-14-00147-f002:**
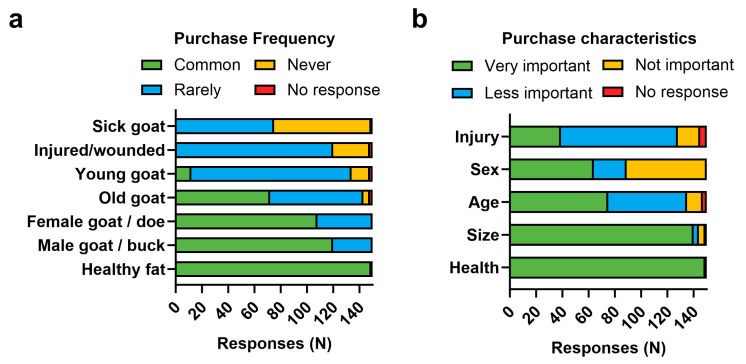
**Characteristics of goats at market**. (**a**) Frequency of characteristics from goats purchased by butchers (*n* = 150), (**b**) importance of different characteristics in purchase-making decisions (*n* = 150).

**Table 1 animals-14-00147-t001:** **Commonality of plastic and hardware IFBs identified by butchers**. Percent of butchers who find plastic or hardware IFBs and whether these are listed as ‘not common’ or a ‘major problem’ from the survey.

District	*n*	Plastic IFBs			Hardware IFBs		
		Not Common	Major Problem	Total	Not Common	Major Problem	Total
**Chitipa**	18	8 (44.4%)	6 (33.3%)	14 (77.8%)	3 (16.7%)	0 (0%)	3 (16.7%)
**Lilongwe**	50	46 (92%)	4 (8%)	50 (100%)	38 (76%)	0 (0%)	38 (76%)
**Nsanje**	50	32 (64%)	0 (0%)	32 (64%)	25 (50%)	0 (0%)	25 (50%)
**Salima**	20	14 (70%)	1 (5%)	15 (75%)	2 (10%)	0 (0%)	2 (10%)
**Thyolo**	12	6 (50%)	4 (33.3%)	10 (83%)	0 (0%)	0 (0%)	0 (0%)
** *Total* **	*150*	*106 (70.7%)*	*15 (10%)*	*121 (80.7%)*	*68 (45.3%)*	*0 (0%)*	*68 (45.3%)*

**Table 2 animals-14-00147-t002:** **Plastic and hardware IFB prevalence among market butchers in different Malawi districts**. * Derived from the ‘last 5 goats’ question, multiplying respondents by 5. † ‘Any’ answers > 0 per respondent, ‘Average’ = the mean from respondents, ‘Total’ = the tally of all goats across respondents.

District	*n*	Goats *	Plastic in the Last Five Goats… †			Hardware in the Last Five Goats… †		
			Any	Average	Total	Any	Average	Total
**Chitipa**	18	90	10 (55.5%)	0.8	14 (15.5%)	2 (11.1%)	0.1	2 (2.2%)
**Lilongwe**	50	250	50 (100%)	2.6	128 (51.2%)	32 (64%)	0.9	43 (17.2%)
**Nsanje**	50	250	12 (24%)	0.5	25 (10%)	3 (6%)	0.2	9 (3.6%)
**Salima**	20	100	6 (30%)	0.6	11 (11%)	1 (5%)	0.1	1 (1%)
**Thyolo**	12	60	7 (58.3%)	1.2	14 (23.3%)	0 (0%)	0	0 (0%)
** *Total* **	*150*	*750*	*85 (56.7%)*	*1.3 (1.4)*	*192 (25.6%)*	*38 (25.7%)*	*0.4 (0.7)*	*55 (7.3%)*

**Table 3 animals-14-00147-t003:** Butchers’ comments on the impact of plastic IFBs on meat quality.

Comment on Plastic IFB Impact	Tally
Carcass weighs less than expected	3
Damaged intestines	1
Meat has little or no fat	9
Meat is tough	2
Meat colour is different	2
Meat flavour is different	1
Meat is harder to clean and sell	1
No differences noticed	72

## Data Availability

Original anonymised tabulated data are provided in Appendix A, blank copies of questionnaires are available upon request.

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
