# Peer review of "Prevalence of Plastic and Hardware Foreign Bodies among Goats at Malawi Markets"

_animals, 2024, doi:10.3390/ani14010147_

Round 1

Reviewer 1 Report

Comments and Suggestions for Authors

The manuscript describes the prevalence of indigestible foreign bodies (IFBs) in the goats from 5 districts from Malawi. Data were collected by a survey in 150 market stand butchers and it distinguish plastic IFBs and hardware IFBs. Prevalence was quite high, particularly for plastic IFBS (80.7%). These data are interesting to concern veterinarians and consumers about the importance of anthropogenic plastic pollution in the environment, which may cause damage not only to wildlife but also to domestic animals such as goats.  Some aspect to improve the manuscript are:

1.- Authors state that survey was answered by 150 market butchers, however, IFBs are detected when the forestomach is opened, which occurs in the slaughterhouse. Thus, authors should state surveys was conducted on slaughterhouses, although they are small or familiar.

2.- If possible, it would be interesting to include a picture of an IFBs.

3.- In abstract it is stated that “Butchers commented that goats containing plastic IFBs typically had less fat or weighed less than 24 expected”, however, in Table 3 only carcass with IFBs weight less than expected and 9 carcass with IFBs had less fat than expected while 72 carcass with IFBs had no changes. Consider rewriting the sentence in abstract to avoid this discrepancy.

4.- Since sick animals typically are not sold to go to the slaughterhouse, it is possible that an important number of goats with IFBs suffering important clinical sign such as impaction, internal haemorrhages, etc. have not been included in the survey because they do not go to the slaughterhouse. Consider including a sentence in discussion to state this.

Author Response

Please find attached our responses and thank you for your time reviewing this manuscript.

Reviewer 2 Report

Comments and Suggestions for Authors

Dear authors,

thank you for sending this interesting manuscript concerning the occurrence of plastic and hard materials in the gut of goats after slaughter and their possible impact on animal health and meat quality.

I think the study was done properly, but some details are missing in the methods part and also in the discussion. Please try to clarify them to bring this manuscript in an acceptable form.

Introduction: Please formulate a specific research question or hypothesis to make even more clear what the aim of the study was.

Methods:

Describe the eligibility criteria for the butchers.

Please describe development and validation of the questionnaire. How was the questionnaire distributed? By mail, or direct approach at the markets? Did they fill out the questionnaire on their own or together with some researcher from the study?

Can you explain why you invited 150 butchers and not less or more? What did you do to avoid selection bias?

Please explain which statistical test was used for which hypothesis. Did you examine subgroups and if yes, how did you select them? How did you deal with missing data?

Results

L. 156: How did you define young and old?

Table 1 and 2: Confidence intervals around the percentages could help to assess if there were differences between the regions, if this is an interesting question.

Please report actual p-values instead of p<xxx.

l. 210 f. Please rephrase this sentence. "Stochasticity" os mentioned 2 times, this is a bit confusing.

ll. 217 ff: I am sure that it is complicated to describe, and I am not sure if I understood it completely: More butchers reported that they have spotted IFBs in goats (23 instead of 12 -> almost doubled) - probably a sign of increased awareness after the first survey. But, the number of goats with plastic increased only slightly from 25 to 30 -> less animals per butcher, right? This results in an increase to 0.65 animals per 5 animals, which is plausible to me. The next sentence is not clear for me. You distinguish between plastic and hardware, and in both cases, there were significantly more observations compared to the first survey? But then I don't understand why you summarize that not more butchers find IFBs (there are more) and the overall IFBs did not change (they did, even statistically significant in McNemars test). Could you clarify?

l. 227 ff: Why did only these 50 give information? And did they give free answers or were these answers predefined in a list?

And what did you mean with "impacted slaughter"? The slaughter process or rather the quality of the meat or the health status of the animal?

A lot of these questions should be addressed in the methods section, so take care where to place the information.

Discussion

The discussion is well-written, but in my opinion, the results themselves are not discussed enough while the conclusions part focusing on the avoidance or ban of plastics is a bit too large. Since you did not study the effect of different plastic "concentration" or if goats avoid plastic uptake if enough nourishment is available, you should not put too much emphasis on this point.

In addition, I miss the discussion concerning the butchers who were asked twice - there are many interesting things to discover and to discuss, I think. Obviously, the awareness has increased a lot -> that could mean that in the first survey, there was a lot of underestimation due to missing awareness. What could that mean for the whole study?

Then, I miss the discussion of possible limitations of the study design (voluntary participation? sometimes answers refused, etc.) and of possible selection bias. Also, only univariable analyses were performed. Which influence factors might have played a role as well?

Author Response

Please find attached our responses to your review and thank you fro your time and insightful comments.

Round 2

Reviewer 2 Report

Comments and Suggestions for Authors

Dear authors,

thank you for addressing all my concerns. I am happy now with the manuscript and will recommend acceptance.